# First Report on Reproductive Features of Shadow Driftfish *Cubiceps whiteleggii* (Perciformes: Nomeidae): An Effort toward Sustainable Management

Jun Ohtomi [1] , Kaito Hirowatari [2], Md Mosaddequr Rahman [2,*] , Lindon Havimana [2,3] and Yasuji Masuda [1]

[1] Faculty of Fisheries, Kagoshima University, 4-50-20 Shimoarata, Kagoshima 890-0056, Japan; ohtomi@fish.kagoshima-u.ac.jp (J.O.); k2217151@kadai.jp (Y.M.)

[2] Graduate School of Agriculture, Forestry and Fisheries, Kagoshima University, 1-21-24 Korimoto, Kagoshima 890-0065, Japan; kaito.hirowatari@gmail.com (K.H.); lindon.havimana@sinu.edu.sb (L.H.)

[3] Department of Fisheries Studies, Solomon Islands National University, Honiara P.O. Box R113, Solomon Islands

* Correspondence: k8457926@kadai.jp or mosaddeq.bd@gmail.com; Tel.: +81-99-286-4111; Fax: +81-99-286-4015

**Abstract:** Fish of the Nomeidae family are a significant part of marine food webs and are an emerging fisheries resource. The formulation of sustainable management and conservation strategies is constrained by the lack of comprehensive research on the population biology of any of these species. We, therefore, studied reproductive aspects of *Cubiceps whiteleggii* as a representative of Nomeidae. We used a biological index (gonadosomatic index), as well as macroscopic (visual observations of the surface and transverse ovary sections) and microscopic analyses, in search of proxies for estimating the ovarian maturity status. Samples were collected from April 2014 to December 2017 in Kagoshima Bay, Japan. Asynchronous ovary development in *C. whiteleggii* indicated multiple spawning during a single reproductive season. Ovarian maturity stages were divided into six groups based on the appearance of the most advanced oocytes, post-ovulatory follicles, and atretic oocytes: immature, maturing, mature, spawned, spent, and resting. Females with ovaries at mature or spawned stages were defined as sexually mature individuals. The size at 50% sexual maturity was estimated to be 126 mm standard length. The spawning season was estimated to last from July to November. Both biological and macroscopic indices were found unsuitable to be used as independent indices in estimating reproductive aspects requiring scientists and managers to search for better alternatives. The conceptualization of appropriate management protocols for the fishery should be aided by these findings, which are the sole knowledge available on the reproductive features of this species. This will ensure its sustainable harvest and make it easier to conduct future research on fish reproduction.

**Keywords:** reproduction; oocyte development; ovarian maturation; macroscopic index; microscopic index; sexual maturity; spawning season; sustainable management; Nomeidae




## 1. Introduction

Fishes of the family Nomeidae constitute an important component of marine food webs in pelagic ecosystems [1]. Members of this family are caught as bycatch in large numbers by long-liners and tuna purse seiners [2–4]. Species of the genus *Cubiceps* have been found in the stomachs of top predators such as tunas, swordfish, seabirds, fur seals [5–12], dolphins [13–15], and balaenopterid whales [3]. Despite their wide distributions and immense ecological significance, little has been done to elucidate different aspects of the population biology of any species of this family that are important for their sustainable management and conservation. Fish reproduction plays a vital role in population productivity and its resilience to fishing pressures and environmental changes [16]. Hence, there is a need to understand the reproductive biology of species of this family to ensure their sustainable harvest and management.

The species we have selected in this study is *Cubiceps whiteleggii*, a species of Nomeidae often found in the coastal waters of Japan [17]. The benthopelagic fish *C. whiteleggii* inhabits depths between 150 and 550 m [17,18] and is widely dispersed in warm-temperate and tropical regions of the Indian Ocean (apart from the Red Sea) and ranges in the Indo-western Pacific from Japan to central-eastern Australia [17–19]. This species is caught in reasonable quantities off southwestern India and the Philippines [18]. In the coastal waters of Japan, this species is distributed from Hokkaido to Kyushu in the Pacific coast, the Sea of Japan, the East China Sea [17,20], and the Osumi Islands [21]. In recent years, *C. whiteleggii* has gained popularity as an edible fish and its market value has been increasing since, especially in Japan. In the wake of food scarcity worldwide, the demand for seafood is on the rise. We not only need to ensure the sustainability of the existing fisheries resources but also to find out and ensure the sustainable exploitation and management of emerging fisheries resources. As the exploitation of *C. whiteleggii* increases, studies on its reproductive biology that include ovarian maturation, size and age at sexual maturity, and spawning season are necessary for successful fishery management through establishing mesh size regulations and closed season for fishing. However, except for limited descriptions of morphology [17,22,23], genetic characterization [19], and distribution [18,21], there is a significant gap in studies on the population biology (particularly the aforementioned aspects of reproductive biology) of *C. whiteleggii*, much like with other species of the Nomeidae family. Therefore, our study focused on providing the first comprehensive information on the ovarian maturation, size at sexual maturity, and spawning season of *C. whiteleggii*, which will also shed light on the reproductive nature of fishes of other nomeid species.

In this study, we also planned to clarify another important issue in fish reproductive studies. Biological indices and macroscopic analyses are often used as simple and cost-effective methods of determining the gonadal maturity status and, subsequently, the spawning season of fish. In most cases, these indices are, however, not verified through more sophisticated methods of estimating ovarian maturation, such as histological observation of the gonads. Lack of verification might lead to erroneous determination of the maturity status and subsequent reproductive aspects. Ambiguous determination of reproductive aspects of fish (such as maturity status and spawning season) and their use in policymaking remain a major shortcoming in achieving sustainability in fisheries management. Therefore, we tested the suitability of the most commonly used biological index, the gonadosomatic index (GSI), as well as a macroscopic index in determining the spawning season of *C. whiteleggii*, hypothesizing that these indices are suitable enough to be used as simple and reliable methods in estimating different reproductive aspects of fish. The findings presented here will be important for creating and putting into practice conservation strategies, for the effective management of the species, and for expanding our understanding of this understudied family. Additionally, the results will expand our understanding of the suitability of using different biological and macroscopic indices in fish reproductive studies.

## 2. Materials and Methods

### 2.1. Study Site

Our study site, Kagoshima Bay, located approximately at 31°25′ N, 130°38′ E in southern Kyushu, Japan, is a deep semi-enclosed bay with water depths exceeding 230 m (Figure 1). The bay consists of two submarine craters connected by a shallow channel with an opening at its southern end facing the Pacific Ocean. The bay is subjected to the warm Kuroshio Current [24], and bottom water changes during the winter season [25]. The bay is home to a large number of fish and shellfish species that are commercially important and support a commercial fishery [26].

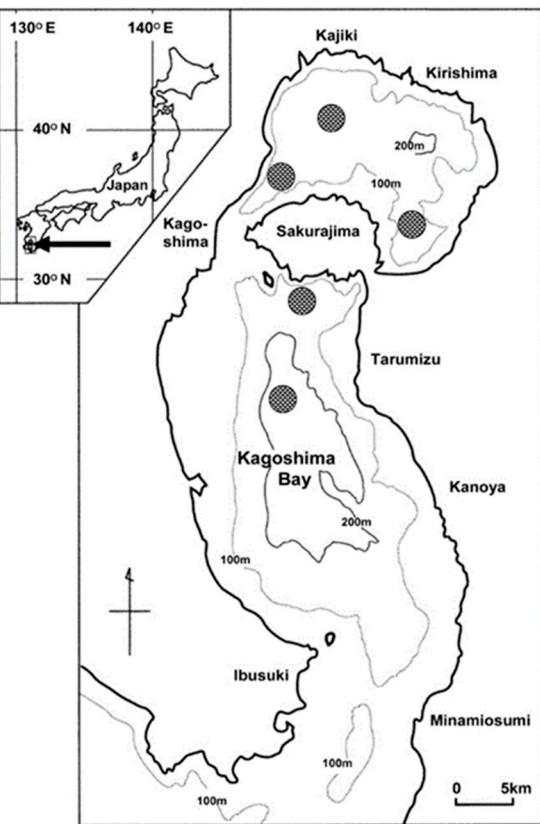

**Figure 1.** Sampling locations (shaded) where *Cubiceps whiteleggii* were caught in Kagoshima Bay, Japan.

### 2.2. Sampling

Experimental trawl surveys were done monthly at depths between 130 and 230 m to collect *C. whiteleggii* samples. (Figure 1). The samplings were conducted on board the training vessel Nansei-Maru (175 t) of the Faculty of Fisheries, Kagoshima University from April 2014 to December 2017 using a simple trawl net carrying canvas kites on the tips of the wings (LC-VI; Nichimo Corporation, Tokyo, Japan). The net was 23.5 m long and 8 m wide, with a mesh size of 37.9 mm and 20.2 mm at the net body and cod end, respectively [27].

Using a compact temperature-depth logger (ATD-HR; JFE Advantech, Nishinomiya, Japan) attached to the net's head rope, the net depth and water temperature were measured at 1 min intervals. The water depth was recorded at 1 min intervals with a split-beam quantitative echo sounder (KFC-3000; Kaijo Corp., Hamura, Japan). The net was towed at 2 knots for the predetermined 10 min duration. The bottom water temperatures recorded during effective tow durations at the sampling sites were used to calculate monthly water temperature. According to the logger-echo sounder depth plots, the effective tow duration was calculated from the first and last times the net made contact with the ocean floor for each tow [28].

Individuals of *C. whiteleggii* caught from each haul were counted and chilled immediately in ice. The samples were subsequently brought to the laboratory the same day and fixed in 10% formalin for further examination. Additional samples caught by a commercial fishing boat using a small seine net (mesh size: 27.5 mm and 23.3 mm at the net body and cod end, respectively) from January 2016 to December 2017 were also used. A total of 1275 individuals (789 males and 486 females) were collected. Only females were used in the present study.

### 2.3. Measurements and Macroscopic Observation

For each individual, standard length (SL) was measured on a measuring board to the nearest 1 mm, body weight (BW) was measured to the nearest 0.01 g on an electronic balance (UX6200H, Shimadzu Corp., Kyoto, Japan), and ovarian weight (OW) was measured to the nearest 0.001 g on an electronic balance (UX620H, Shimadzu Corp., Kyoto, Japan). The GSI was calculated as: GSI (%) = $(100 \times OW)/((BW - OW))$. Thereafter, the ovaries were visually observed and classified into different stages based on the visibility and appearance of oocytes on the surface and transverse ovary sections (see Results, Section 3.1).

### 2.4. Histological Analysis and Measurement of Oocytes

To determine the level of ovarian maturation, small samples of tissue from the center portions of the 364 ovaries were removed and histologically analyzed. The tissues were cut into sections of 4 to 6 µm thickness, fixed in paraffin, and stained with Mayer's hematoxylin and eosin. The developmental stages of oocytes were classified according to Yamamoto [29], Granada et al. [30], and Havimana et al. [31] with some adjustments for this species, and the histological classification of atretic oocytes and post-ovulatory follicles followed Hunter and Macewicz [32]. The presence of the most advanced oocytes, post-ovulatory follicles, and atretic oocytes served as indicators of ovarian maturity estimation [30,31,33–35]. Oocytes that had been sectioned through the nucleus were measured under a microscope to determine their sizes. The majority of the oocytes were not precisely spherical owing to natural variance and histological processing. To limit variance and prevent unnaturally increasing the overlap between oocytes at various developmental stages, average values of the minimum and maximum oocyte dimensions were used (see West [36] and Farrell et al. [37]).

### 2.5. Size at Sexual Maturity

The SL at which 50% of females become sexually mature ($SL_{50}$) was defined as the size at sexual maturity [38]. Females with mature or spawned ovaries were considered sexually mature (see below). Using the software DeltaGraph 7 (Red Rock Software, Salt Lake City, USA), the proportion of sexually mature females (PMF) was plotted versus SL to estimate the size at sexual maturity by fitting a logistic function based on the least-squares approach. The logistic equation was expressed as: $PMF = 100/[1 + Exp(-r(SL_m - SL_{50}))]$, where r is the slope of the curve and $SL_m$ is the median value of each SL class [38].

### 2.6. Annual Reproductive Cycle

The annual reproductive cycle was analyzed based on the monthly percentage occurrence of each ovarian maturity stage determined histologically. Furthermore, the annual reproductive cycle was also estimated using GSI and the monthly occurrence of each macroscopically determined ovarian maturity stage for comparing the results with that of the histological output. Fish specimens smaller than the size at sexual maturity were eliminated from this investigation to reduce the inclusion of immature fish.

## 3. Results

### 3.1. Macroscopic Stages of Ovaries

Based on the visibility and appearance of oocytes on the surface and transverse sections, the ovaries of *C. whiteleggii* could be distinguished into three macroscopic stages (Figure 2). Stage A ovaries had invisible oocytes on both the surface and the cross-sections. Stage B ovaries had oocytes invisible and individually indistinctive on the surface but visible in the cross-section, whereas Stage C ovaries had visible and individually distinctive oocytes on both the surface and cross-sections. Stage C ovaries were considered mature and the appearance of females with Stage C ovaries was considered to indicate spawning season.

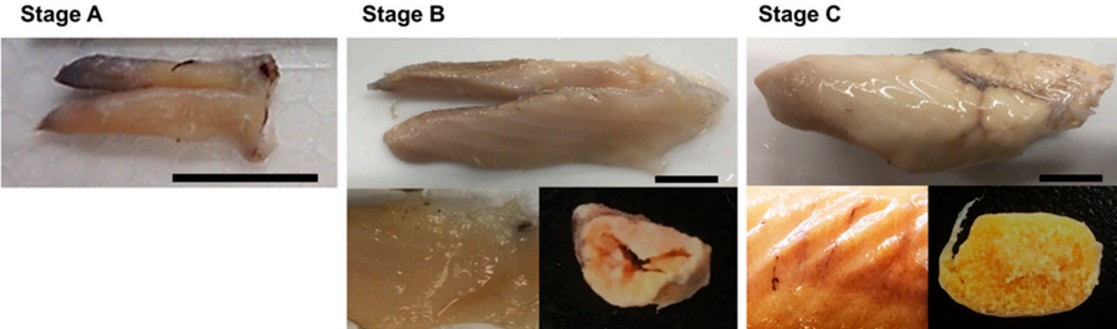

**Figure 2.** Macroscopic maturity stages of ovaries of *Cubiceps whiteleggii*, surface only (Stage **A**), surface with the transverse section (Stage **B**), and surface with transverse section (Stage **C**). Scale 1 cm.

### 3.2. Classification of Oocyte Developmental Stages

This study observed the following developmental stages of the oocytes during histological analysis of ovaries: peri-nucleolus, previtellogenic, early yolk globule, late yolk globule, migratory nucleus, hydrated, early atretic, late atretic, and post-ovulatory follicle. In the peri-nucleolus stage (Figure 3a), the nucleus was relatively small, and multiple nucleoli of different sizes could be found in its periphery. The cytoplasm was basophilic and stained dark blue by hematoxylin. Oocytes were ≥70 μm in diameter. In the previtellogenic stage (Figure 3b), the nucleus was relatively larger and multiple nucleoli could be seen around its periphery. The cytoplasm was basophilic and oocyte diameter ranged from 60 to 120 μm. In the early yolk globule stage (Figure 3c), small acidophilic yolk globules started to appear in the cytoplasm. Oocyte diameter ranged from 110 to 150 μm. In the late yolk globule stage (Figure 3d), yolk globules increased in size and covered the whole cytoplasm. Oil droplets around the nucleus increased in size. Oocyte diameter ranged from 140 to 280 μm. In the migratory nucleus stage (Figure 3e), the nucleus migrated toward the animal pole where smaller yolk globules occurred and the yolk globules began to coalesce. Oocyte diameter ranged from 260 to 310 μm. In the hydrated stage (Figure 3f), yolk globules completely underwent coalescence and formed a single translucent yolk mass. Oocyte diameter ranged from 260 to 410 μm. In the early atretic stage (Figure 3g), the yolk globules started to disintegrate and reabsorbed with follicular layers. This stage corresponds with the alpha stage atresia described by [32]. In the late atretic stage (Figure 3h), the follicle decreased in size and was characterized by numerous disorganized granulosa cells and intracellular vacuoles. This stage corresponds with the beta stage atresia described by [32]. Additionally, the post-ovulatory follicle (Figure 3i), which was characterized by a granulosa cell layer, coiled up within a less irregular thecal cell layer covering, and was frequently observed during the spawning season.

### 3.3. Classification of Ovarian Maturity Stages

*C. whiteleggii* had asynchronous ovaries, containing oocytes at various developmental stages i.e., from migratory nucleus to post-ovulatory follicles. The stages of ovarian maturity were thus divided into six categories based on the occurrence of the most advanced oocytes, post-ovulatory follicles, and atretic oocytes: immature, maturing, mature, spawned, spent, and resting (Table 1). Ovaries with unyolked oocytes only, such as peri-nucleolus and/or previtellogenic stages, were classified as immature. Maturing ovaries contained early and/or late yolk granule staged oocytes. The most advanced oocyte stage in mature ovaries was either migratory nucleus or hydrated. Spawned ovaries contained yolked oocytes in addition to postovulatory follicles. In case of spent ovaries, more than 50% of the oocytes were in their early atretic stage; however, postovulatory follicles were absent. Resting ovaries contained late atretic staged oocytes, as well as unyolked oocytes.

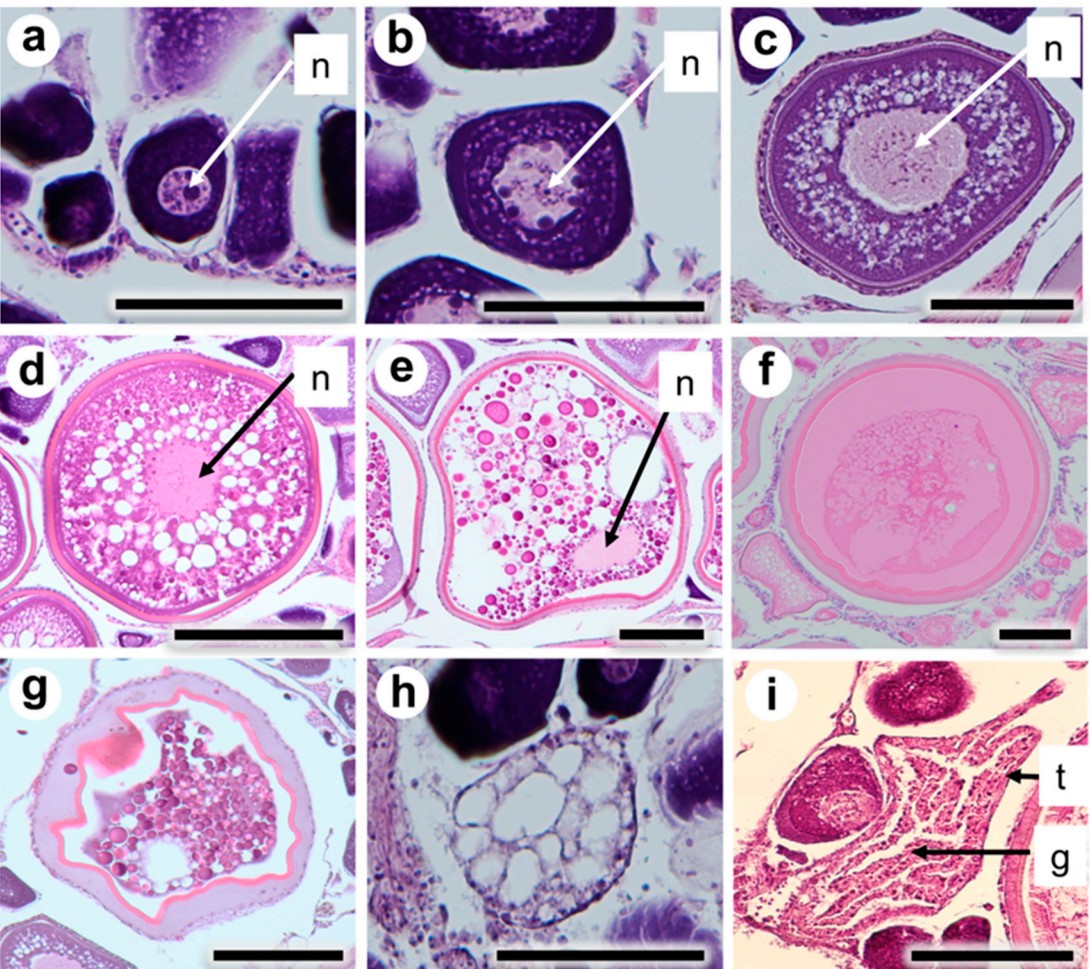

**Figure 3.** Oocyte developmental stages and post-ovulatory follicle of female *Cubiceps whiteleggii*: (**a**) peri-nucleolus, (**b**) previtellogenic, (**c**) early yolk globule, (**d**) late yolk globule, (**e**) migratory nucleus, (**f**) hydrated follicle, (**g**) early atretic follicle, (**h**) late atretic follicle, and (**i**) post-ovulatory follicle. n Nucleus, g granulosa cell layer, t thecal cell layer. Scale bars 100 μm.

**Table 1.** Histological characteristics and diameter of oocytes in ovaries at different stages of maturity in female *Cubiceps whiteleggii*.

| Ovarian Maturity Stage | Developmental Stage of Oocytes |
|---|---|
| Immature | Only unyolked oocytes, such as PN and PV, were present. Oocytes ranged from 30 to 120 μm in diameter. |
| Maturing | The most advanced oocyte was in EYG or LYG. Oocytes were 110 to 280 μm in diameter |
| Mature | The most advanced oocyte was in MN or H. Oocytes were 260 to 410 μm in diameter |
| Spawned | Yolked oocytes and POF were present |
| Spent | More than 50% of yollked oocytes were in EA. POF was absent |
| Resting | Unyolked oocytes and LA oocytes were present |

PN, peri-nucleolus stage; PV, previtellogenic stage; EY, early yolk globule stage; LYG, late yolk globule stage; MN, migratory nucleus stage; H, hydrated stage; EA, early atretic stage; LA, late atretic stage; POF, postovulatory follicle.

### 3.4. Size at Sexual Maturity

The size at sexual maturity was estimated from the 82 females collected from July to September when mature females began to appear and peaked (see Results, Section 3.5). The

SL of females collected in the present study ranged from 66 mm to 171 mm. The minimum size of mature females was 119 mm SL. The estimated logistic function was as follows: (PMF = 100)/([1 + Exp (−0.087 (SL$_m$ − 126)))] (n = 11, r$^2$ = 0.977). Size at 50% maturity (SL$_{50}$) was estimated to be 126 mm (Figure 4). The relationship between SL and GSI is shown in Figure 5. The GSI values rose sharply from around 110 mm in SL, indicating the probable size at sexual maturity.

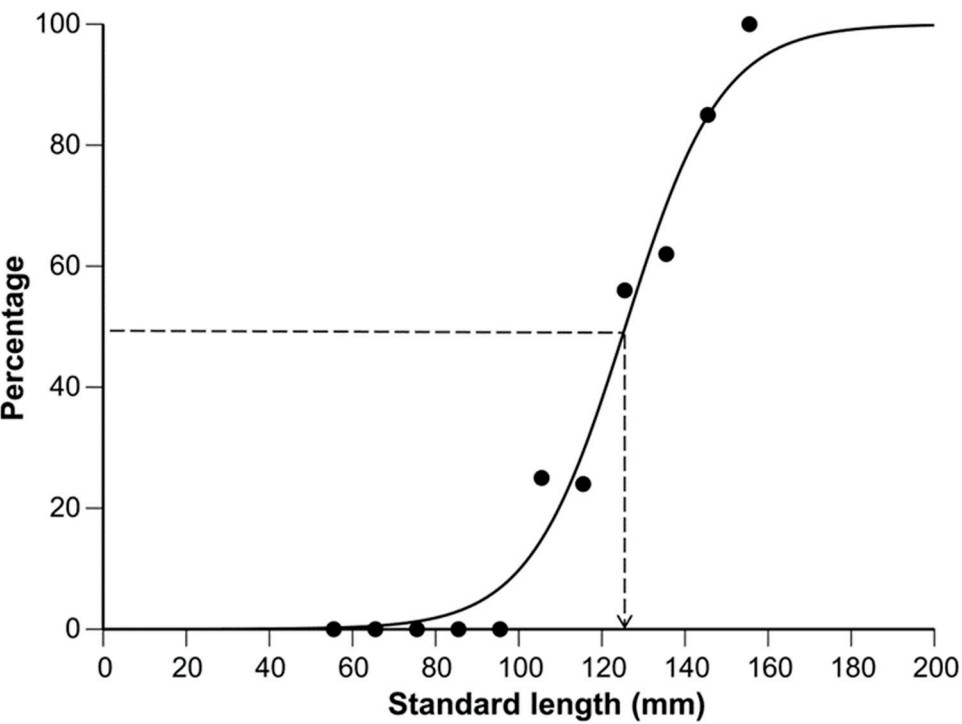

**Figure 4.** Percentage of sexually mature females of *Cubiceps whiteleggii* in Kagoshima Bay, Japan at 10 mm standard length (SL) intervals. A logistics curve was fitted to the data and the dotted arrow shows the SL at 50% maturity.

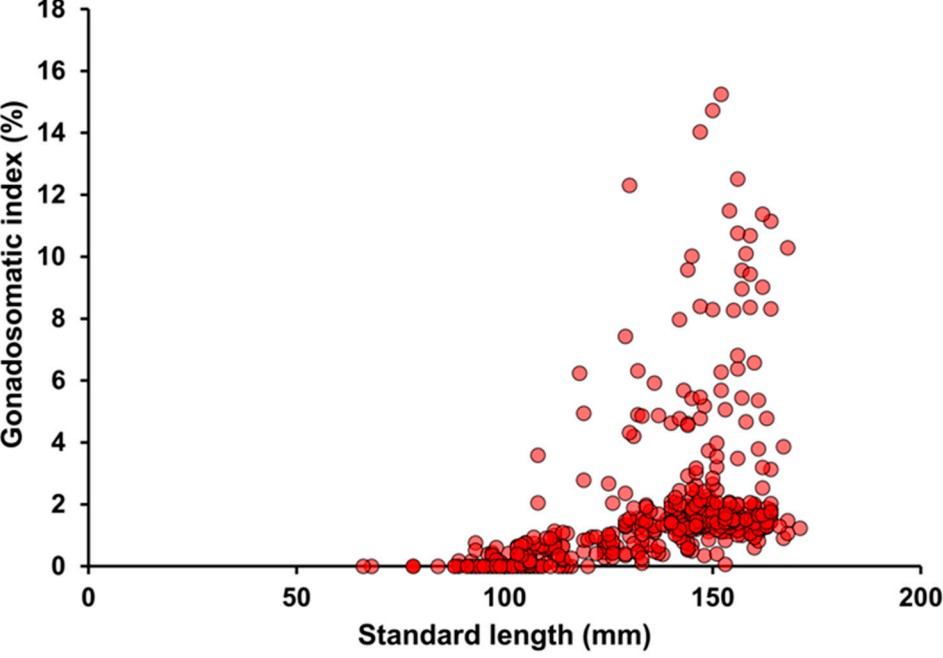

**Figure 5.** Relationship between the standard length and gonadosomatic index of female *Cubiceps whiteleggii*.

### 3.5. Annual Reproductive Cycle

According to the monthly ratio of the six stages of ovarian maturity, females were typically immature from January to April, developing from May to June, mature or spawned from July to November, and spent from around October to November (Figure 6). Therefore, the spawning season of *C. whiteleggii* was deemed to last from July to November.

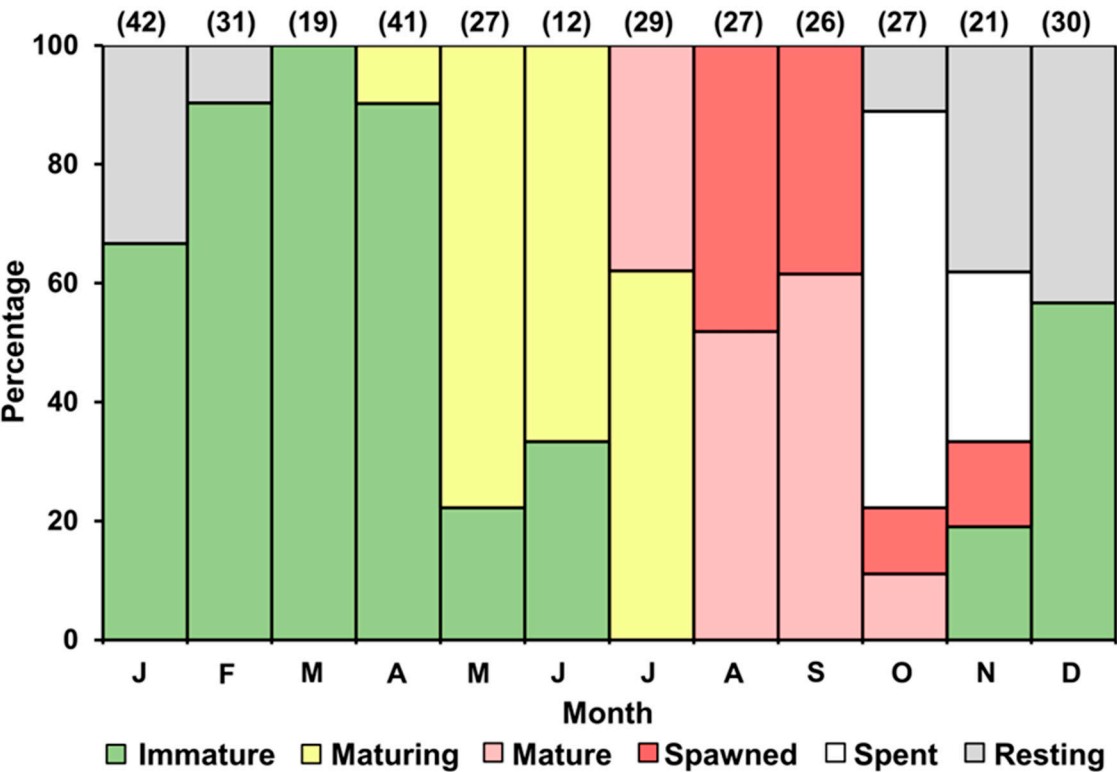

**Figure 6.** Percentage occurrence of each ovarian maturity stages of female *Cubiceps whiteleggii* in Kagoshima Bay, Japan. Fish smaller than the size at 50% maturity (<126 mm in standard length) were excluded from this analysis. The number of females examined is given in parentheses.

The monthly changes in GSI showed that GSI values started to increase around June before experiencing a rapid increase in July. A similar abrupt decline in mean GSI value was recorded in September before reaching a minimum point in December (Figure 7). This suggested that the probable spawning season of *C. whiteleggii* lasted from around July to around September. The percentage occurrence of each ovarian maturity stage determined through macroscopic analysis showed that females with Stage C ovaries tended to occur from May to October. While the remaining months were dominated by females with Stage A and Stage B ovaries (Figure 8). This indicated that the probable spawning season of *C. whiteleggii* lasted from May to October.

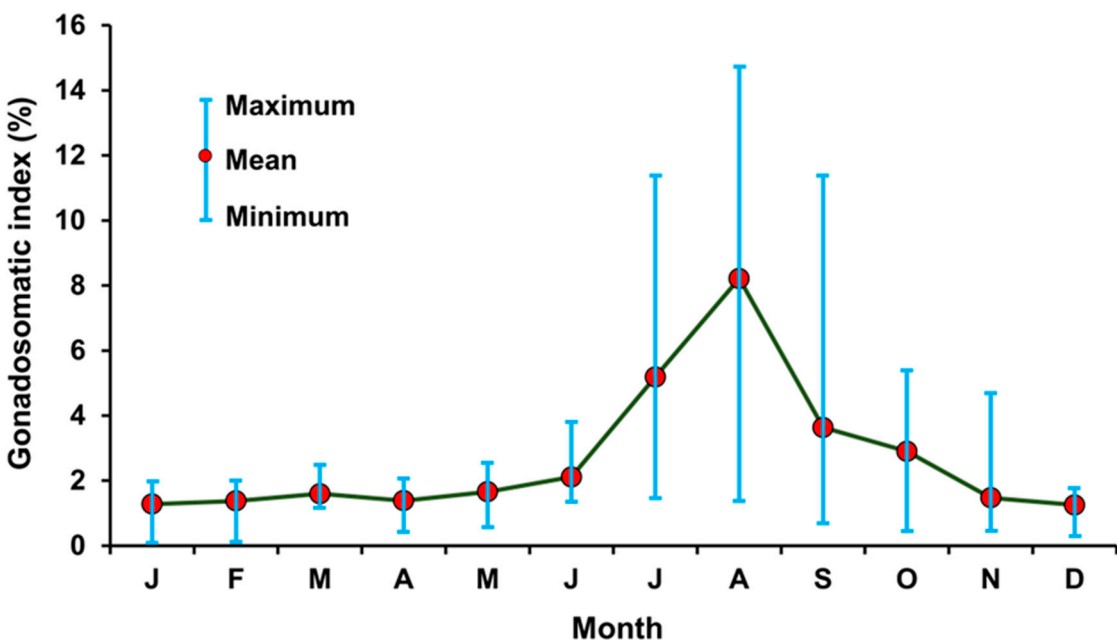

**Figure 7.** Monthly changes in gonadosomatic index of female *Cubiceps whiteleggii* in Kagoshima Bay, Japan.

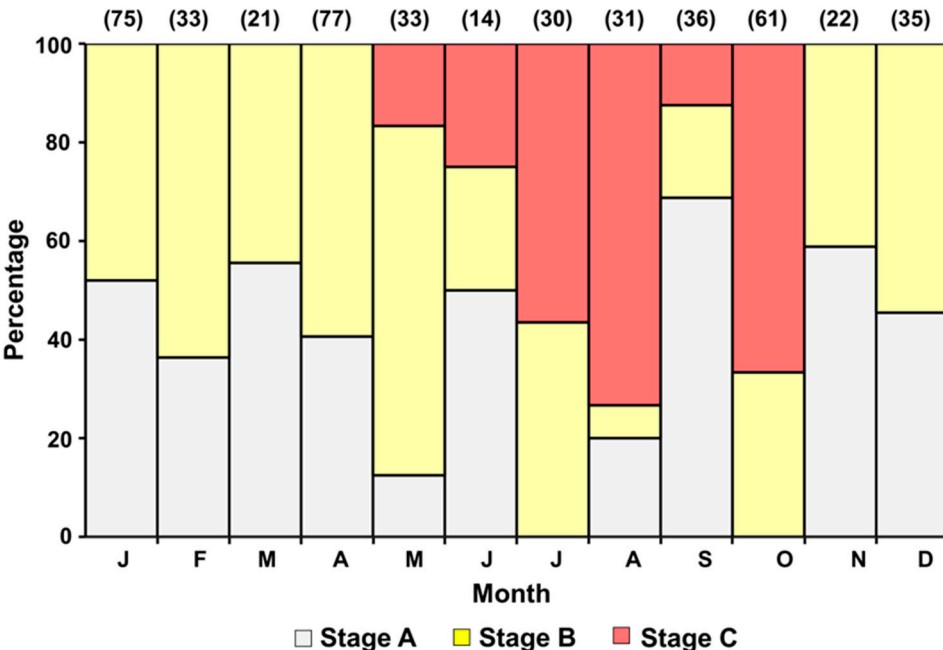

**Figure 8.** Percentage occurrence of the three ovarian maturity stages of female *Cubiceps whiteleggii*. The number of females examined is given in parentheses.

*3.6. Bottom Water Temperature*

The monthly bottom water temperature in the sampling area ranged from 14.9 °C in March to 15.9 °C in January (Figure 9). The mean bottom water temperature in Kagoshima Bay fluctuated seasonally within a range of 1 °C. A steady gradual increase in mean bottom water temperature was noticed from July before experiencing a slight decrease in December followed by the highest recorded value in January.

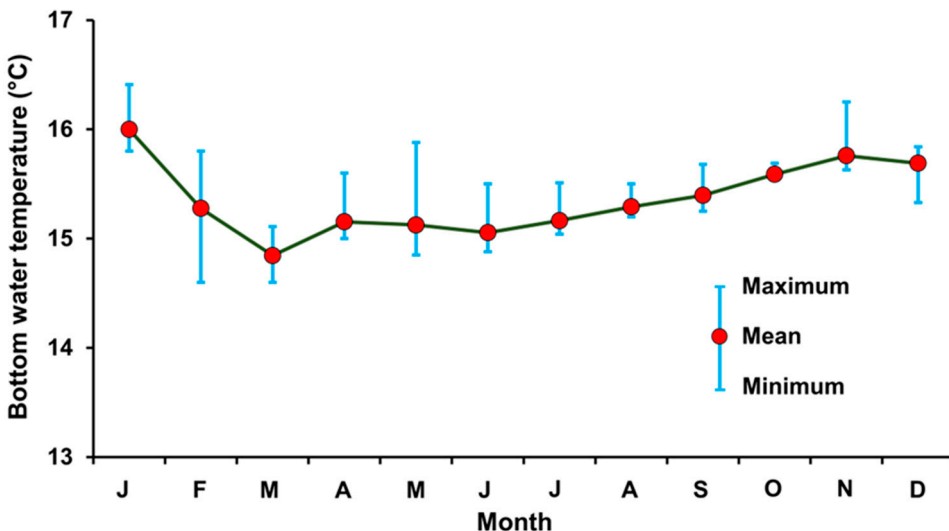

**Figure 9.** Monthly changes in bottom water temperature at the sampling where mature females were caught in Kagoshima Bay, Japan from January 2015 to December 2017.

## 4. Discussion

Information regarding the population biology of the members of the family Nomeidae is scarce. We conducted a pioneering reproductive study of *C. whiteleggii* as a representative of Nomeidae. The evaluation and management of exploited fish stocks depend on an accurate assessment of fish maturity conditions [39–42], so we used a histological approach to determine the ovarian maturity status of the ovary. This method is the most accurate method currently available for evaluating the fish's maturity state [36,39,40]. *C. whiteleggii* showed an asynchronous ovary, a phenomenon also reported in other fish species of different taxonomic families including those from the similar geographic location such as *Apogon lineatus* [43], *Nemipterus bathybius* [30], *Gerres equulus* [44], and *Evynnis tumifrons* [31]. The ovarian maturity stages of *C. whiteleggii* were therefore classified based on the appearance of the most advanced oocytes, post-ovulatory follicles (POF), and atretic oocytes into six categories (immature, maturing, mature, spawned, spent, and resting). Females with oocytes in their migratory nucleus or hydrated stages and females with POF in their ovaries (i.e., mature, or spawned ovaries) were considered sexually matured individuals. It is well-known that a migrating nucleus and hydrated oocytes suggest impending spawning [32], while post-ovulatory follicles constitute histological evidence of imminent spawning [45], despite the temperature dependence of the maturation process [46]. In some species, such as *Pagrus major* and *Dentex hypselosomus*, nucleus migration begins approximately 24 h ahead of spawning, and the following oocyte hydration process begins approximately 12 h ahead of spawning, while POFs that occur at ovulation are probably reabsorbed within 24 h [47,48]. Thus, these stages of *C. whiteleggii* were thought to emerge a few days after spawning. The existence of POFs and asynchronous oocyte development indicates that repeated spawning will occur during a reproductive season [49,50], which ensures the production of more eggs and, therefore, juveniles than a single spawning [51]. Species with asynchronous ovarian development require special attention while planning for their sustainable management because such species tend to have a large size-at-age range [52]. Fish of a particular species may mature at different times during a spawning season, or even in separate years, depending on their size at age [52]. This wide range of sizes at age also affects when fish first become exposed to capture [53]. Understanding a fish's size at sexual maturity is imperative for successful management [39,54]. Based on the histological examination of the ovary, our study determined that the minimum size of mature females is 119 mm SL and that $SL_{50}$ is 126 mm. Based on the relationship between GSI and SL, however, the size at sexual maturity was estimated to be around 110 mm SL. A similar discrepancy in the estimated size at sexual maturity was also reported for

*Evynnis tumifrons*. While Havimana et al. [31] evaluated the minimum size of mature female *E. tumifrons* as 161 mm fork length (FL) and $FL_{50}$ as 179 mm, based on the histological approach, Anzawa et al. [55] calculated the minimum size of mature female *E. tumifrons* as 170 mm FL from its GSI. It has been demonstrated that the histological method is more accurate for estimating size at maturity [33,56]. In accordance with the recommendation of [31], we thereby suggest using histological techniques to estimate size at sexual maturity.

Sexually matured females of *C. whiteleggii* started to appear in July, indicating the initiation of their spawning season. The conclusion of spawning in November was marked by the occurrence of fish with spent ovaries, which indicates the imminent cessation of spawning [30–32,57]. Even though the rapid increase in mean GSI in July followed by a similar decline in September might tempt us to conclude that this species has a spawning season from around July to September, histological observation of ovarian maturity status proved this to be misleading.

The relationship between GSI and histological ovarian maturation progression was, however, positive (Figure 10). There was a significant increase in the percentage occurrence of mature females with a GSI class between 4% and 5% (nearly 75%) over that of the earlier GSI classes of 3% to 4% (nearly 20%) and 2% to 3% (nearly 10%). More than 95% of females with GSI $\geq$ 5% were found to be sexually mature. Females with GSI $\geq$ 4% could therefore be considered sexually mature. Females with GSI $\geq$ 4% occurred from June to November, indicating the probable spawning season, which approximated the histological outcome (Figure 7). Similarly, the precision of size at sexual maturity estimation increased significantly as well. GSI $\geq$ 4% started to appear in females having an SL of 120 cm, which approximates the minimum size at sexual maturity assessed by histological observation. These outcomes indicate that GSI used together with histological data could provide clarity in understanding the reproductive aspects of fish, but it should not be used alone as an independent index for management decision-making prior to histological verification.

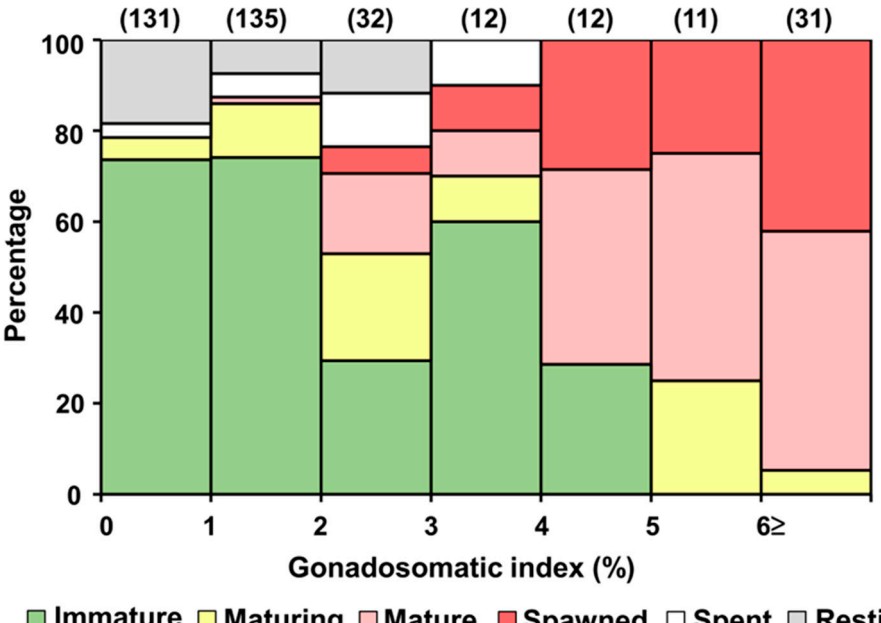

**Figure 10.** Percentage occurrence of histologically determined ovarian maturity stages against associated gonadosomatic index class of female *Cubiceps whiteleggii* in Kagoshima Bay, Japan. The number of females examined is given in parentheses.

The output of the probable spawning season based on macroscopic analysis (May to October), however, was not similar to the figure estimated by histological analysis. This difference could be attributed to the lack of ability to detect hydrated oocytes by macroscopic analysis, and this signified the limitation of the method in the determination of mature females. Ovarian stage classification based on macroscopic analysis of transverse sections

of the ovary and visible surface grains on fish eggs was therefore deemed unsuitable for determining reproductive aspects of fish. In accordance with [36], we highly recommend that, in any macroscopic determination of maturity or biological index, histology should be conducted simultaneously to validate the accuracy of information gathered through this method.

It is well established that day length and temperature are two of the most significant environmental cues that can give a reliable signal for the timing of reproductive and gonadal activities [58]. With a relatively constant food availability in Kagoshima Bay [59], temperature and day length could play a vital role in the initiation and termination of spawning activities. In this study, the bottom water temperature started to increase in April, which coincides with the appearance of maturing individuals. *C. whiteleggii* continued to spawn as the bottom water temperature kept increasing steadily until reaching its seasonal highest point in November. A slight decrease in temperature was noticed in December and, for the first time in six months, there was no spawning activity recorded. Even though the fluctuation in bottom water temperature in Kagoshima Bay is minimal, these small fluctuations seemed to have a significant impact on the initiation and termination of spawning. Thermal inhibition of reproduction is a widespread phenomenon that affects a variety of taxa, habitats, and temperature ranges with the key difference across species is the absolute temperature at which the suppressive effects manifest [60]. While the thresholds for these effects differ in proportion to particular thermal tolerance ranges, all species are likely to exhibit comparable responses to temperature rises. Reproductive aspects especially the timing of sexual maturity and maturation process of several species are also known to be influenced by photoperiod [61]. In rainbow trout, for example, both constant long days and continuous light advance spawning [62]. Long days were found to be most suitable for producing egg and reducing inter-spawning intervals in *Oreochromis niloticus* [63] and *Betta splendens* [64]. Prolonged photoperiod, however, was interestingly found to negatively affect the growth and reproductive performances of *Labeo rohita* [65]. In our study area, the longest average monthly day length was recorded in June and the lowest was recorded in December (Figure 11). Spawning of *C. whiteleggii* was initiated just after the day length was the longest and ceased when the day length was the shortest. The combined effect of temperature and day length varied significantly depending on the geographical location of the study area. *E. tumifrons* spawning in northern Japan (Niigata) was favored when conditions combined warmer water with longer days, while in southern Japan (Fukuoka), a combination of warmer water conditions with shorter days was preferred [31]. Finally, in the southernmost location off the southwestern coast of Japan (Kagoshima), day length did not seem to have much effect [31]. It is, however, impossible for us to have in-depth knowledge of the geographical impact on reproductive aspects of this species.

The seasonality in the reproductive cycle of *C. whiteleggii* could therefore be attributed to fluctuations in bottom water temperature and/or day length, along with other environmental factors. Owing to the understanding of reproductive characteristics of *C. whiteleggii*, fishery managers can safeguard breeding stock by developing fishery enclosures aimed at large spawning aggregations. As a result, a significant portion of the breeding stock could spawn [39,40,67], which will enhance the reproductive output of this emerging fisheries resource and enable their sustainable management and conservation. To develop a more comprehensive understanding of the population's reproductive capacity, which will serve as the foundation for effective management regimes, additional research on spawning frequency and fecundity, as well as an assessment of the breeding stock, are recommended. Because the population biology of the species of Nomeidae has been inadequately studied, our study will act as a platform for future studies of other nomeid species. Furthermore, the clarification of shortcomings in using macroscopic and biological indices in fish reproductive studies will help scientists and managers in management decision-making and searching for better alternatives.

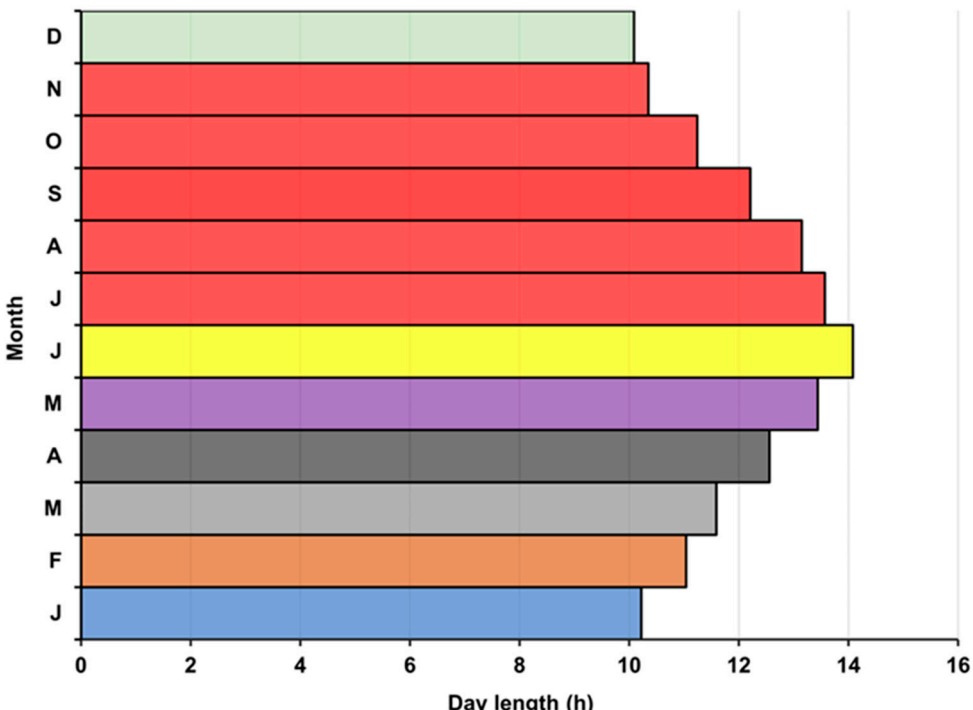

**Figure 11.** Average monthly day length in Kagoshima City, Japan. Cited from the database of the National Oceanic and Atmospheric Administration [66]. Red bars show the months when spawning of *Cubiceps whiteleggii* occurred.

**Author Contributions:** Conceptualization, J.O. and Y.M.; methodology, J.O., Y.M. and K.H.; formal analysis, K.H., L.H. and M.M.R.; writing—original draft preparation, M.M.R. and L.H.; writing—review and editing, M.M.R. and J.O.; supervision, J.O. and Y.M.; funding acquisition, J.O. All authors have read and agreed to the published version of the manuscript.

**Funding:** This work was supported in part by the Japan Society for the Promotion of Science (19K02297 and 22K02161) secured by J.O.

**Informed Consent Statement:** Not applicable.

**Data Availability Statement:** The data presented in this study are available on request.

**Acknowledgments:** We appreciate the assistance of the Nansei Maru training vessel crew from Kagoshima University's Faculty of Fisheries throughout the experimental trawl surveys. We thank Miyuki Ohse of the Tarumizu City Fishermen's Cooperative for donating specimens and the students of the Laboratory of Fisheries Biology, Faculty of Fisheries, Kagoshima University for their unwavering support throughout the research.

**Conflicts of Interest:** The authors declare no conflict of interest.

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
