# Peer review of "First Report on Reproductive Features of Shadow Driftfish Cubiceps whiteleggii (Perciformes: Nomeidae): An Effort toward Sustainable Management"

_sustainability, doi:10.3390/su14148813_

Round 1

Reviewer 1 Report

Please find the attachment below.

Author Response

Thank you for kindly reviewing our manuscript. We have provided our detailed response in the attached file. Please be kind enough to have a look.

Thank you so much once again.

With best regards,

Dr. Md Mosaddequr Rahman

Reviewer 2 Report

I found this work to be a well-presented paper describing the reproductive biology of a fish from the family Nomeidae in Japan. The introduction provided adequate background information and the objectives of the study were clear. The methods section was generally thorough, although I note some specific comments below. Some text that could be beneficial in the methods section is provided in the results and discussion. For example, the histological descriptions read like methods describing how ovaries/oocytes were classified and the results section should be reserved for descriptions of findings from macroscopic and microscopic examinations. The title indicates this work moves management of the population toward sustainability, but the paper does not examine harvest regulations or some other relevant management or conservation goals and how this work can promote sustainability. I suggest deleting the portion of the title “:an effort towards sustainable management”. It seems to over-sell the value of the work without providing direct evidence that this work informs sustainable management. Further, the paper presents this work as informing management of species across a taxonomic family, but many families I have worked with have considerable diversity in life history. How well does the study species represent other fishes in the family Nomeidae? The authors may consider reducing the amount of text indicating this work could inform management of an entire taxonomic family. The authors note that day length and water temperature likely influenced gonadal development and reproductive processes, but do not provide a formal analysis examining these environmental factors on spawning or ovary stage. The authors provide some descriptive evidence but a statistical analysis may help support their claims. The information on reproductive biology of this species appear to fill an important knowledge gap for the species, but it is unclear how applicable this information is to other species in the family and how specifically this study will directly improve sustainability. The findings related to GSI and histological analysis appear to be beneficial for other scientists considering tradeoffs associated with estimating maturity using different approaches.

Specific Comments

Line 11: Delete “the” in “the marine food webs”

Line 16: The text in parentheses is long. Consider shortening or omitting the text in parentheses.

Line 18-19: It might be helpful for readers if this sentence defined whether this was for individual fish or the population.

Line 22-23: If spawning is asynchronous, can we reliably define maturity status?

Line 24-26: This reads to me like the authors are stating the methods they used were not suitable to meet the study objectives. If I am interpreting this correctly, can the results be trusted?

Line 46-51: The authors might consider adding a map of the species range to the manuscript.

Line 53-54: Is there a reference to support this sentence?

Line 56-58: Reproductive biology studies may be important, but this sentence does not directly link the science to management action or monitoring. How exactly will this work inform successful management?

Line 64-65: Is this appropriate? How do we know that information from this species would applicable to other species?

Line 72-75: The authors might consider providing a reference or example.

Line 97-103: How were sampling sites selected?

Line 108-110: How frequently were trawl tows conducted? If trawl tows only took place a few days per month, using the temperatures from those tows as a monthly index of water temperature may not provide an adequate index.

Line 115-116: The authors might consider providing comparable levels of detail from the small seine net as is provided for the trawl survey to permit readers to understand the appropriateness  of combing the datasets.

Line 117: Is the total number of individuals here applicable to the seine net or the seine and trawl in aggregate?

Line 124-126: Define the stages for readers unfamiliar with this sort of work. Which of the stages in Figure 2  would be considered mature?

Line 128: The authors note having 486 female fish above. Were samples from 122 fish not examined?

Line 148-150: The authors might consider providing some references for the form of the logistic curve that they used. Also, the authors do not provide details on the size classes used (i.e., what are the bounds of the size classes?). What is the benefit of using size classes versus observed standard lengths?

Line 156-157: How was the size at sexual maturity determined? Was this SL50? If so, 50% of the mature fish could be excluded from analysis. Also, if all fish were assigned a maturity status histologically, could the authors only use fish that exhibited characteristics of maturation and avoid using length to segregate mature and immature fish?

Line 214-215: This is a considerable reduction from the overall sample size presented earlier in the paper. Please explain why the sample size was reduced for analysis.

Figure 5: It could be argued that once fish reach sizes typical of maturity that length does not provide a good predictor of GSI given the variability in GSI shown. The authors might consider using different colors to show months or seasons when fish were collected to potentially help explain some of the variation shown here.

Line 242: C. whiteleggii is not italicized here but has been elsewhere. Check for consistency throughout.

Line 273-275: The authors have “This section is not mandatory but can be added to the manuscript if the discussion is unusually long or complex.” which seems to be guidance text from the publisher.

Line 275-277: The authors might consider noting these species are from different families if applicable. It may be beneficial to note taxonomic families for all fishes described to clarify which are in the same family as the study species.

Line 280-282: This text reads like methodological details to me. The authors might consider moving this sentence to the methods.

Author Response

Thank you for kindly reviewing our manuscript. Your comments were constructive and educational. We have provided our detailed revision in the attached file. Please be kind enough to have a look.

Thank you so much once again.

With best regards,

Dr. Md Mosaddequr Rahman

Reviewer 3 Report

Sustainability-1798991, entitled "First report on reproductive features of shadow driftfish Cubiceps whiteleggii (Perciformes: Nomeidae): an effort towards sustainable management" analyzes the reproductive seasonality of the species of interest while comparing resulting estimates among differing approaches.  From my review, I find the paper quite well written, relatively easy to follow, and conclusions consistent with the results.  I did find some areas for major improvement, which I have identified in the accompanying document.  In addition, I do need to note that the study does seem to comprise a small portion of the overall distribution of the species, and as such, seems to provide geographically limited information.  Thus, I question whether this work is submitted to the appropriate outlet.  However, that is a decision most appropriate for the editors.

Best wishes for the publication of your work.

Author Response

(The authors gave the same response as above.)
